# RoGeR v3.0.5 – a process-based hydrological toolbox model in Python

Robin Schwemmle[1], Hannes Leistert[1], Andreas Steinbrich[1], Markus Weiler[1]

[1]Hydrology, Faculty of Environment and Natural Resources, University of Freiburg, Freiburg, Germany

*Correspondence to*: Robin Schwemmle (robin.schwemmle@hydrology.uni-freiburg.de)

**Abstract.** Although water availability and water quality are equally important for an effective water resources management at various spatial and temporal scales, to date, a combined representation of soil water balance components and water quality components in Python are not available. The new RoGeR toolbox contains models that can be used for the quantification of hydrological processes, fluxes and stores, but also solute transport

processes based on StorAge selection. This study presents the code structure and functionalities of RoGeR developed as a scientific model toolbox following defined open-source software guidelines. RoGeR uses five different computational back-ends covering just-in-time compilation, parallelism and graphical-processing units that might be used for optimizing computational performance. We show that graphical-processing unit computing has the greatest potential to improve computation time and energy usage, especially for large modelling

experiments. A simple modelling experiment highlights the capabilities of the new RoGeR model toolbox. We simulated the soil water balance, stable water isotope ($^{18}$O) transport and corresponding travel time distributions of the Eberbaechle catchment, Germany, for a three-year period. Due to the current limitations for a variety of process components further development of RoGeR as a scientific software is needed. Future modifications are easily possible due to the open software architecture of RoGeR.

## 1 Introduction

The interplay between the water and solute mass balance (e.g. oxygen-18, chloride or nitrate) and its related flow and transport in the soil-vegetation-atmosphere interface plays an important role for the understanding of hydrologic systems (e.g. Benettin et al., 2017). However, measurements of their states and fluxes are neither in space nor in time ubiquitously available (Beven, 2011). Thus, soil hydrological models, soil-vegetation-

atmosphere-transfer (SVAT) models and distributed catchment models are indispensable tools to complement measurements (e.g. for a better process understanding) and to make predictions (e.g. future climate impacts, land cover changes or in ungauged catchments). Currently, there are many models in hydrology and the landscape of models is highly diverse (from simple conceptual models to complex physically-based models). One reason for this large and diverse landscape of models is that hydrologist still disagree about modelling concepts (Weiler and

Beven, 2015). Despite the large number of models, however, there is a lack of reproducibility in computational hydrology (Hutton et al., 2016; Reinecke et al., 2022). The main reasons for this lack of reproducibility are poorly documented codes and workflows, code being too complex, unavailable code, missing input data, a lack of calibration standards and a lack of standards dealing with uncertainties (e.g. Reinecke et al., 2022).

Simulating the hydrological processes at the soil-vegetation-atmosphere interface including solute mass balance and transport with a high spatial and/or temporal resolution still requires long computation time. For reasons of computational performance, hydrological models such as HYDRUS (Šimůnek et al., 2016), EcH2O-iso (Kuppel et al., 2018a) or mHM (Heße et al., 2017; Kumar et al., 2020; Kumar et al., 2013; Samaniego et al., 2010) are written in low-level programming languages such as Fortran or C++. However, these languages are hard to read, to learn and are usually not included in the curriculum of hydrology-related degree programs. By contrast, high-level programming languages are easier to read and to learn, but computation takes about 3-5 times longer than equivalent code in low-level programming languages (Häfner et al., 2021). Therefore, high-level programming languages have the potential to foster reproducibility. Recently, high-level open-source programming languages such as R or Python gained popularity in the hydrological modelling community. Especially Python is currently the most popular programming language among software users (e.g. IEEE Spectrum, 2022; PYPL, 2022; Stack Overflow, 2021). Hydrological models quantifying the hydrological cycle that are written in Python, for example, are SUPERFLEX (Dal Molin et al., 2021; Fenicia et al., 2011), CWatM (Burek et al., 2020) and UniFHy (Hallouin et al., 2022) but none of these models consider transport of solutes, and they generally focus at the catchment scale. To date, only rsas (Harman, 2015) implemented a solute transport model written in Python. However, rsas does not quantify the water balance and requires hydrological fluxes as input.

For reasons of longer computation times, high-level programming languages are often avoided in spatially distributed hydrological models. One solution to reduce computation time in high-level programming languages is using a just-in-time compiler (JIT). However, Python does not contain a built-in JIT. Instead, Python requires program libraries such as Numba (Lam et al., 2022) or JAX (Bradbury et al., 2018). However, Numba and JAX provide the opportunity to run the code on graphical processing units (GPUs) to decrease computation times. Veros (Häfner et al., 2018; Häfner et al., 2021), an ocean model written in Python using JAX for acceleration, demonstrated that GPU computations are a competitive alternative to central processing units (CPUs). In addition to that, Häfner et al. (2021) could show that GPU computations save energy.

The first model version of RoGeR had a focus on the event-based runoff generation (Steinbrich et al., 2021). Thereafter, RoGeR has been further developed and by adding a routing scheme, surface runoff and subsurface runoff contributions to flooding events could be explicitly simulated (Steinbrich et al., 2021). Additionally, by considering snow hydrological processes, urban hydrological processes and redistribution processes such as evapotranspiration enabled the estimation of the long-term water balance (Steinbrich et al., 2021). Based on the previous development efforts of the RoGeR model by Weiler (2005), Steinbrich et al. (2016) and Steinbrich et al.

(2021), we reimplemented the process-based hydrological model RoGeR in a modular software architecture (e.g.
different hydrological processes are implemented in separate modules that can be independently modified) written
in Python. Since RoGeR had no implementation for solute transport so far, we include solute transport based on
StorAge selection (SAS) functions (e.g. Benettin et al., 2017). We choose a high-level programming language and
a modular software paradigm to foster reproducibility and wide-range application in teaching and research. In
particular, we aim to facilitate general code understanding, writing new code and debugging code which usually
takes most of the time within software projects. To overcome limitations on computational performance, we
include the program library JAX.

EcH2O-iso provides simulations with a spatial resolution of 30 m (Kuppel et al., 2018b) and mHM uses a spatial
discretization that starts at 100 m (Yang et al., 2018). HYDRUS can be used at the plot scale (e.g. Asadollahi et
al., 2020), but applying HYDRUS at the hillslope or catchment scale requires a commercial license. Thus, the
main objective of the toolbox is to provide simulations with a high spatial (<25 m) and temporal resolution (> 10
minutes) at the local scale (< 10 km). The local scale covers different scales: the plot scale (1 – 25 m; Blöschl and
Sivapalan, 1995), the hillslope scale (25 – 100 m; Blöschl and Sivapalan, 1995) and the smaller catchment scale
(100m – 10 km; Blöschl and Sivapalan, 1995). RoGeR contributes to the modelling of the soil-vegetation-
atmosphere continuum by enabling simulations with a fine spatial resolution.

In the following, we describe the implementation of the new model developed as a scientific software following
open-source guidelines. Thereafter, we provide a brief overview about the representation of the hydrological
processes and the related solute transport. We further profile the computational performance and energy usage.
Finally, we demonstrate the capabilities of the model by simulating a three-year period for a synthetic site.

## 2 Implementation

### 2.1 RoGeR as a scientific open-source software

For the development of RoGeR as scientific open-source software, we followed the guidelines presented in Table
1. We defined these guidelines based on van Gompel et al. (2016) and Hall et al. (2022), and on reviewing earth
science related software written in Python (e.g. Bakker et al., 2016; Bartos, 2020; Burek et al., 2020; Collenteur
et al., 2019; Dal Molin et al., 2021; Häfner et al., 2018; Hallouin et al., 2022; Helmus and Collis, 2016; Kratzert
et al., 2022; Mälicke, 2022; May et al., 2022; Rose, 2018; Schwemmle et al., 2021). We organised the different
software concepts by the degree of complexity. The application of different software concepts incorporates
different workload and require different knowledge in computer science. In order to keep the workload at a
minimum level, we organised the application by the degree of software complexity. An example for a high degree
of complexity could be a complex model or data analysis tool. Such software is often shipped with many lines of
programming code. In order to maintain the software and make it reusable, we suggest a range of different software

concepts (see Table 1). Hands-on-tutorials, online documentation and unit tests may consume extra working hours and increase the workload of a single researcher, but in the long-term workload of other researchers can be reduced by facilitating the usage of existing methodology. An example for using existing methodology could be a simple application script (e.g. running the model or analyse the data) which contains often a few lines of code and hence

the complexity is low. If existing methodology is used to generate scientific results and no new functionalities are added, comments, public access and a license are sufficient for reusing the approach. Moreover, including these guidelines in the curriculum of hydrology-related degree programs may lay the foundation for a reproducible future in computational hydrology.

### 2.1.1 Software architecture

The basic modular structure of the software is adapted from Häfner et al. (2018). The core modules implement hydrological processes and solute transport. As such, these modules represent a toolbox which can be used to build pre-defined models (e.g. a SVAT model by considering only vertical processes). We already provided some pre-defined models, but in general new models can be easily assembled and combined to the level of complexity that is required. Moreover, further processes might be added by writing new modules. In addition to that, further

modules are available for the pre-/post-processing, writing the model output and handling computational back-ends. RoGeR is pure Python, hence not all computational bottlenecks might be solvable. In such cases, we recommend writing extensions using Cython instead of using a low-level language which would require a compiler.

### 2.1.2 Computational back-ends

The computations are handled by five different back-ends which are implemented through a function decorator (Häfner et al., 2018). Users have to choose a suitable backend beforehand. The choice depends on programming skills, size of the modelling experiment and available computational resources. In the following, we briefly describe the back-ends and give recommendations on the usage:

- numpy: This back-end uses NumPy (Harris et al., 2020) for computation and, hence, is easy to use.

120         However, the interpreted execution of the code and running computations on a single CPU may cause performance limitations. We recommend this back-end to beginners and for small-scale modelling experiments. As long as the modelling experiment fits in the memory, there are no specific requirements for the computational resources.

- numpy-mpi: The numpy-mpi back-end parallelizes the numpy back-end via mpi4py (Dalcin et al., 2011).

125         The size of the modelling experiment might be limited by available memory and number of CPU cores. We recommend this back-end to users with experience in parallelized computations.

**Table 1** Guidelines for scientific open-source software in computational hydrology

| | Term/Concept | Description | Advantage |
|---|---|---|---|
| **Simple complexity** | Comments | Write meaningful and clear comments within the source code | Comprehensive and reusable source code |
| | Public access | Store the source code in an online-repository and use version control to enable collaborative development | Transparent and accessible code |
| | License | Regulates usage of the software | Reusable code |
| **Medium complexity** | Versioning | Assigns a new version number after each release (i.e. update of the source code) | Traceable code |
| | Software environment | Use a virtual environment to avoid software conflicts and provide information about software dependencies | Quick installation |
| | Hands-on-tutorials | Guidance on the application of the software for real case examples | Learnable and lower entry bar for new users |
| **High complexity** | Modules | Individual code blocks | Improves code readability |
| | Logger | Captures errors, warnings and progress of computation | Facilitates debugging of the code |
| | Online documentation | Contains essential information for installing, using the software and theoretical background (e.g. equations) | Learnable and understandable code |
| | Unit tests | Tests basic functionality of the software | Facilitates code maintenance |
| | Continuous integration | Runs unit tests on different operating systems and different software stack | |
| | Profiling | Measures computation time (e.g. of individual modules) or memory usage (e.g. GPU memory) | Supports efficient allocation of computational resources |

-    jax: The jax back-end is the same as numpy, but code is JIT compiled via JAX (Bradbury et al., 2018). Since JAX transforms NumPy code, it is required that all code is NumPy compatible. The JIT compilation leads to decreasing computation time (see Sect. 3).

    -    jax-mpi: Same as numpy-mpi, but code is JIT compiled via mpi4jax (Häfner and Vicentini, 2021). This leads to computational speedup (see Sect. 3).

-    jax-gpu: The code is JIT compiled and computations are performed on GPU which leads to computational speedup (see Sect. 3). The jax-gpu backend requires an appropriate GPU. The size of the modelling experiment is limited to available GPU memory. We recommend this back-end to users with advanced programming skills.

### 2.1.3 Discretization and data handling

For RoGeR-1D (i.e. no lateral transfer between grid cells), space can be represented either through grid cells or polygons. By contrast, RoGeR-2D models (i.e. lateral transfer between grid cells) require a regular grid as spatial representation. In order to generate physically meaningful results, we recommend a spatial resolution between $0.5 \times 0.5$ m$^2$ and $5 \times 5$ m$^2$.

RoGeR requires input data for the following variables:

-    precipitation up to 10 minutes time steps

    -    air temperature at daily time steps

    -    potential evapotranspiration at daily time steps

    -    solute concentrations at daily time steps (only if solute transport is simulated)

The 10 minutes time step is required for the detailed representation of the runoff generation processes (i.e.
infiltration, surface runoff and lateral subsurface runoff). Averaging the input flux for longer time steps leads to an overestimation of infiltration and underestimation of overland flow and preferential flow. Hourly precipitation or daily precipitation datasets can be used with the model and resampled to 10 minutes, however, losing the required temporal variability to correctly simulate the runoff generation processes. For heavy rainfall intensities (the default threshold is > 5 mm/10 minutes), the time step is adapted to 10 minutes (Figure 1). For non-heavy
rainfall intensities (<5 mm/10 minutes and > 0 mm/10 minutes), the simulations use an hourly time-stepping. While no rainfall occurs, a daily time step is used. If precipitation data is available with coarser temporal resolution, for example, hourly or daily resolution, we recommend to resample the precipitation data to the required 10 minutes resolution. Depending on the resolution of the available precipitation data, different resampling methods can be applied. For example, hourly date can be linearly interpolated 10 minutes or daily data
can be disaggregated (e.g. Förster et al., 2016; Koutsoyiannis and Onof, 2001).

The input data can be a time series or spatio-temporal data (i.e. time series for each grid cell) which is either provided as text files (.txt) or NetCDF files (.nc). If the input data is provided as a time series using text files, the data is internally converted to NetCDF.

Metadata (e.g. units, description) for all variables and constants are defined in single modules as dictionaries (Häfner et al., 2018). From these dictionaries, metadata (e.g. units) is automatically added to the model output data. All model output is written to NetCDF files. A major advantage of the NetCDF format is that I-O operations enables parallel writing with compression (Häfner et al., 2018). This reduces time of I-O operations and the size of output files.

## 2.2 Hydrological model

Different hydrological processes are implemented as modules. In the following, we list the already implemented processes and refer to the module and declare whether the module is tested or testing is still ongoing:

- Surface water storage (*surface.py*; testing is complete)
- Soil water storage (*soil.py*; testing is complete)
- Root zone water storage (*root_zone.py*; testing is complete)
- Subsoil water storage (*subsoil.py*; testing is complete)
- Groundwater water storage (*groundwater.py*; testing is ongoing)
- Transpiration (*evapotranspiration.py*; testing is complete)
- Soil evaporation (*evapotranspiration.py*; testing is complete)
- Interception (*interception.py*; testing is complete)
- Snow accumulation/Snow melt (*snow.py*; testing is complete)
- Infiltration driven by capillary forces (*infiltration.py*; testing is complete)
- Infiltration driven by gravitational forces (*film_flow.py*; testing is ongoing)
- Surface runoff (*surface_runoff.py*; testing is ongoing)
- Lateral subsurface runoff (*subsurface_runoff.py*; testing is ongoing)
- Lateral groundwater flow (*groundwater_flow.py*; testing is ongoing)
- Percolation (sub*surface_runoff.py*; testing is complete)
- Capillary rise (*capillary_rise.py*; testing is complete)
- Crop phenology (*crop.py*; testing is ongoing)

The main reason for using a modular structure, is to support the readability of the code. Another motivation for using a modular structure is to represent a certain process by multiple process formulations that provide different complexities (Knoben et al., 2019). As such the processes can be combined in multiple ways to build different model structures. Thus, depending on the chosen process complexity, model structures represent the considered by different degrees of complexity. However, building process-consistent model structures from many different

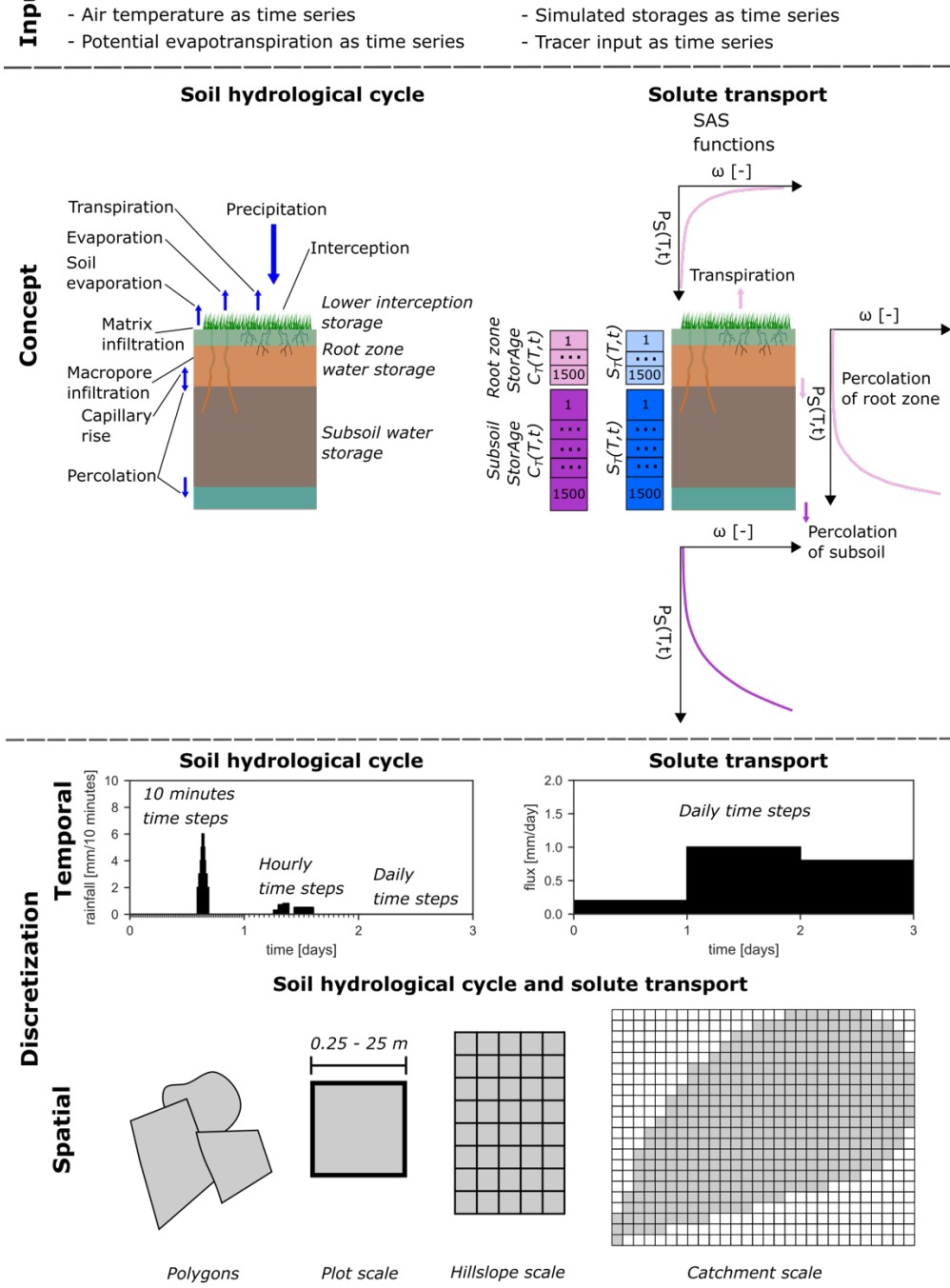

**Figure 1** Overview of model inputs, conceptual implementation shown for a single grid cell or unit (Water storages are represented in italic), and temporal discretization of the soil hydrologic cycle and solute transport. Spatial discretization for different scales is the same for the soil hydrologic cycle and solute transport.

process formulations can be challenging to model users. RoGeR uses single process formulations that constrains the flexibility of the structural complexity. However, we provide pre-defined model structures (i.e. a combination of various hydrological processes) to ensure a certain process consistency. The most basic model structure is shown in Figure 1 and is the basis for more complex model structures. We pre-defined further model structures by adding further hydrological processes (e.g. lateral subsurface flow, crop phenology). For more details about the pre-defined model structures, we refer to the online documentation of RoGeR (Schwemmle, 2023).

RoGeR provides representations for bucket-type interception, degree-day based snow accumulation and snow melt (LARSIM-Entwicklergemeinschaft, 2021), soil matrix, macropore and shrinkage crack infiltration (Steinbrich et al., 2016; Weiler, 2005), soil evaporation (Or et al., 2013), vegetation phenology and vegetation-specific transpiration (Steduto et al., 2009), capillary rise from a groundwater table and percolation to the groundwater (Salvucci, 1993) and lateral subsurface runoff (Steinbrich et al., 2016; Stoll and Weiler, 2010). For detailed information (e.g. model equations), we refer to the online documentation of RoGeR (Schwemmle, 2023).

RoGeR solves explicitly the soil water balance (i.e. fluxes update the state in a specific sequence) using an adaptive time-stepping scheme (see Figure 1). The adaptive time stepping provides a better compromise between accuracy and performance compared to fixed time stepping schemes (Clark and Kavetski, 2010). Numerical errors may compensate for model structural errors, we have not evaluated the effect of other time stepping schemes on the numerical errors of RoGeR. Although numerical errors affect the simulations, parameter uncertainty (e.g. Wagener and Gupta, 2005) or input data uncertainty (e.g. Yatheendradas et al., 2008) may have a stronger impact on the simulations.

## 2.3 Solute transport model

Solute transport is implemented by a travel-time based approach. Particularly, we use StorAge selection (SAS) functions (Rinaldo et al., 2015). SAS is implemented by specific distribution functions. We assign a distribution function to each hydrological process (Figure 1). Here, we introduce two distribution functions which can be used for SAS and are implemented in the toolbox. The first distribution function is based on a power law and requires only a single parameter $k_Q$ (Fig. 2a). The power law distribution function is given as:

$$\omega_Q(T, t) = k_Q^{k_Q} \cdot P_s(T, t)^{(k_Q - 1)}$$

(1)

with

$$P_S(T, t) = \frac{S_T(T, t)}{S(t)}$$

(2)

and the corresponding cumulative power law distribution function:

$$\Omega_Q(T,t) = P_s(T,t)^{k_Q}$$

230        (3)

where $T$ is the water age (day), $t$ is the time step (day), $\omega_Q(T,t)$ is the probability distribution function of the hydrologic flux, $\Omega_Q(T,t)$ is the cumulative probability distribution function), $S_T(T,t)$ is the cumulative age-ranked storage (mm), $S$(t) is the soil water content (mm) and $P_S(T,t)$ is the cumulative probability distribution of the storage.

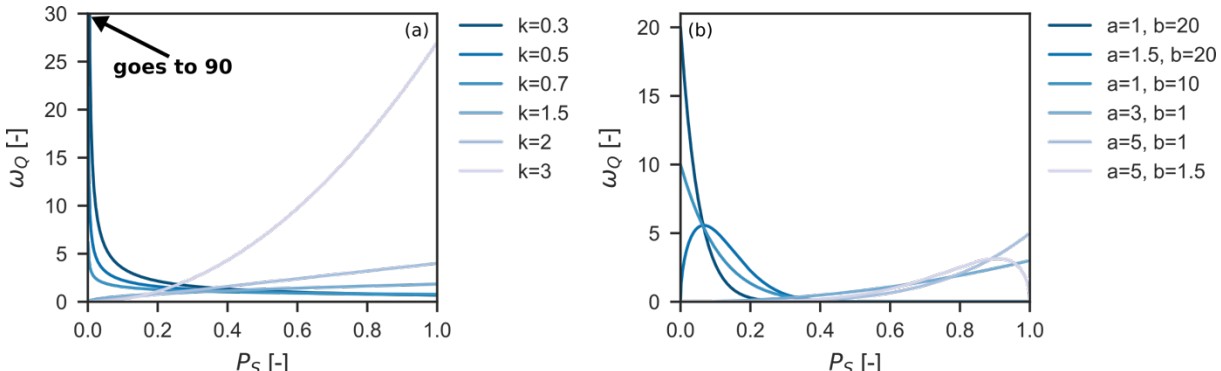

**Figure 2** Storage selection with different parameters illustrated for power law distribution function (a; see Eq. (1)) and Kumaraswamy distribution function (b; see Eq. (3))

As a second distribution function, we employ the Kumaraswamy distribution (Kumaraswamy, 1980). With two parameters $a_Q$ and $b_Q$, the Kumaraswamy distribution provides a greater flexibility than a power law distribution

(Fig. 2b). The Kumaraswamy distribution function is formulated as:

$$\omega_Q(T,t) = a \cdot b \cdot P_s(T,t)^{(a_Q-1)} \cdot \left(1 - P_s(T,t)^{a_Q^{(b_Q-1)}}\right)$$

        (4)

and the corresponding cumulative Kumaraswamy distribution function:

$$\Omega_Q(T,t) = 1 - \left(1 - \left(P_s(T,t)\right)^{a_Q^{b_Q}}\right)$$

245        (5)

Generally, any distribution function might be used as long as a closed form (i.e. probabilities integrates to one) is available (Harman, 2015). We apply the fractional SAS function type (fSAS; van der Velde et al., 2012) and solve the SAS equations for each hydrologic flux $Q$. To solve the SAS functions, we provide three numerical schemes with fixed time steps: (i) deterministic (i.e. solving SAS equations for each flux in a sequential order), (ii) explicit

Euler and (iii) explicit Runge-Kutta fourth-order. Transport processes can be defined for conservative and non-conservative solutes:

-        Stable water isotopes oxygen-18 ($^{18}$O) and deuterium ($^2$H): Isotopic fractionation is not yet considered.

- Bromide and chloride: Evapoconcentration, sorption processes and partitioning of root uptake are included.

- Nitrate: Biogeochemical processes denitrification (Kunkel and Wendland, 2012), nitrification, soil nitrogen mineralization and nitrogen uptake by crops are implemented.

Again, we refer to the online documentation of RoGeR for detailed information (Schwemmle, 2023). The following routines

are implemented, and we refer to the module and declare whether the module

is tested or testing is still ongoing:

- Solute transport and water ages (*transport.py* and *sas.py*; testing is complete)

- Nitrogen cycle (*nitrate.py*; testing is ongoing)

## 3 Test cases for continuous development, computational performance and energy usage

RoGeR uses unit tests and continuous integration to test and ensure technical functionality (see Table 1).
Additionally, we use test cases for continuous development. The idea of these test cases is to guarantee predictive consistency and to track advances in model development (i.e. comparison between model versions). We run the test cases with model parameters that cover a wide range of common parameters and perform simulations with different input data. In contrast to unit tests, the execution time is longer and depends on the number of time steps covered by the input data. The results (see Sect. S1) can be compared to future versions of RoGeR.

**Table 2** Hardware specifications of computational benchmarks

|  | Notebook | Cluster node |
|---|---|---|
| CPU | Intel® Core™ i7 @ 2.60 GHz (four physical cores) | 2 x Intel® Xeon® E5-2680v4 (Broadwell) @ 2.40 GHz (28 physical cores) |
| TDP[1] of CPU | 45 Watt | 280 Watt |
| RAM | 8 GB DDR3 | 128 GB DDR4 |
| GPU | - | Nvidia Tesla K80 (12 GB GDDR5 memory) |
| TDP[1] of GPU | - | 300 Watt |
| Software stack | GNU 8.1, Open MPI 4.1.3, HDF5 1.12.2, roger 3.0 | GNU 9.2, Open MPI 4.1.3, HDF5 1.12, CUDA 11.4, roger 3.0 |
| PUE[2] | 1 | 1.31 |

[1]Total power draw
[2]Power usage efficiency

High-level programming languages such as Python still have the reputation of being comparatively slow. We profiled the computation time and energy usage using the five back-ends (see Sect. 2.1.2). For the profiling, we
used two different hardware specifications representing commonly available computing resources and high-

performance computing (HPC) resources (Table 2). We measured computation time and energy usage with a fixed number of iterations, but varying number of grid cells (Fig. 3).

Model parameters are the same for each grid cell. Figure 3 shows that for small modelling experiments (< 1000 grid cells), the numpy back-end performs equally well as the other back-ends. Parallel computation improves
computational speed up only for intermediate to larger modelling experiments (> 1000 grid cells), provided that a greater number of CPU cores are available. Computation on a single GPU device is faster than on multiple CPUs for the RoGeR-SVAT type model, while multiple CPUs (numpy-mpi and jax-mpi) are faster than a single GPU device for the RoGeR-SVAT-$^{18}$O type model. However, a major requirement for GPU computing is that the modelling experiment fits into the GPU memory (< $10^6$ grid cells). A solution to the memory limitation would be
the usage of multiple GPU devices.

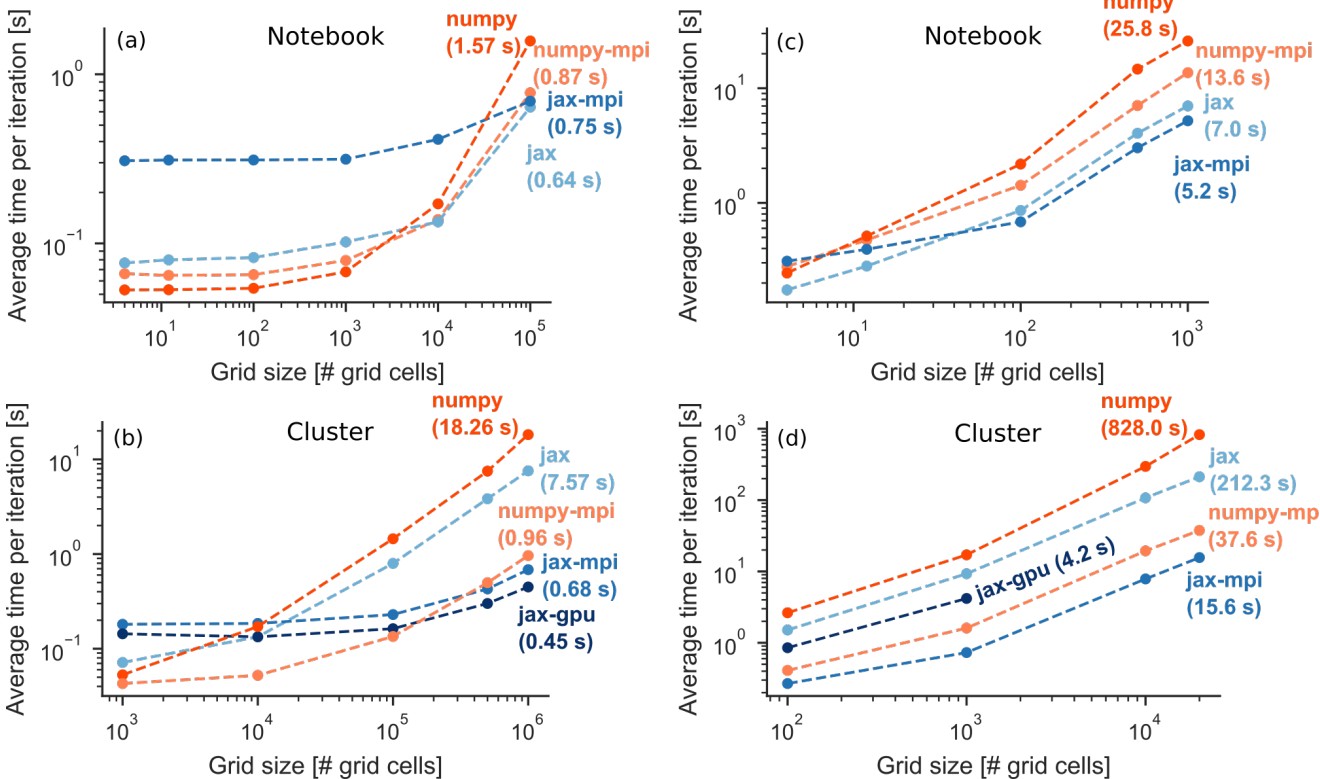

**Figure 3** Runtime performance of computational back-ends for the RoGeR-SVAT type model (a, b) and for the RoGeR-SVAT-$^{18}$O type transport model (c, d). Note that, the number of grid cells represents the two horizontal spatial dimensions (e.g. longitudes and latitudes). The total number of elements is greater for transport models due to additional age dimensions
and can be derived by multiplying the number of grid cells (i.e. two spatial dimensions) with the number of water ages (e.g. 1500). SVAT model used 100 iterations and SVAT-$^{18}$O transport model used 20 iterations.

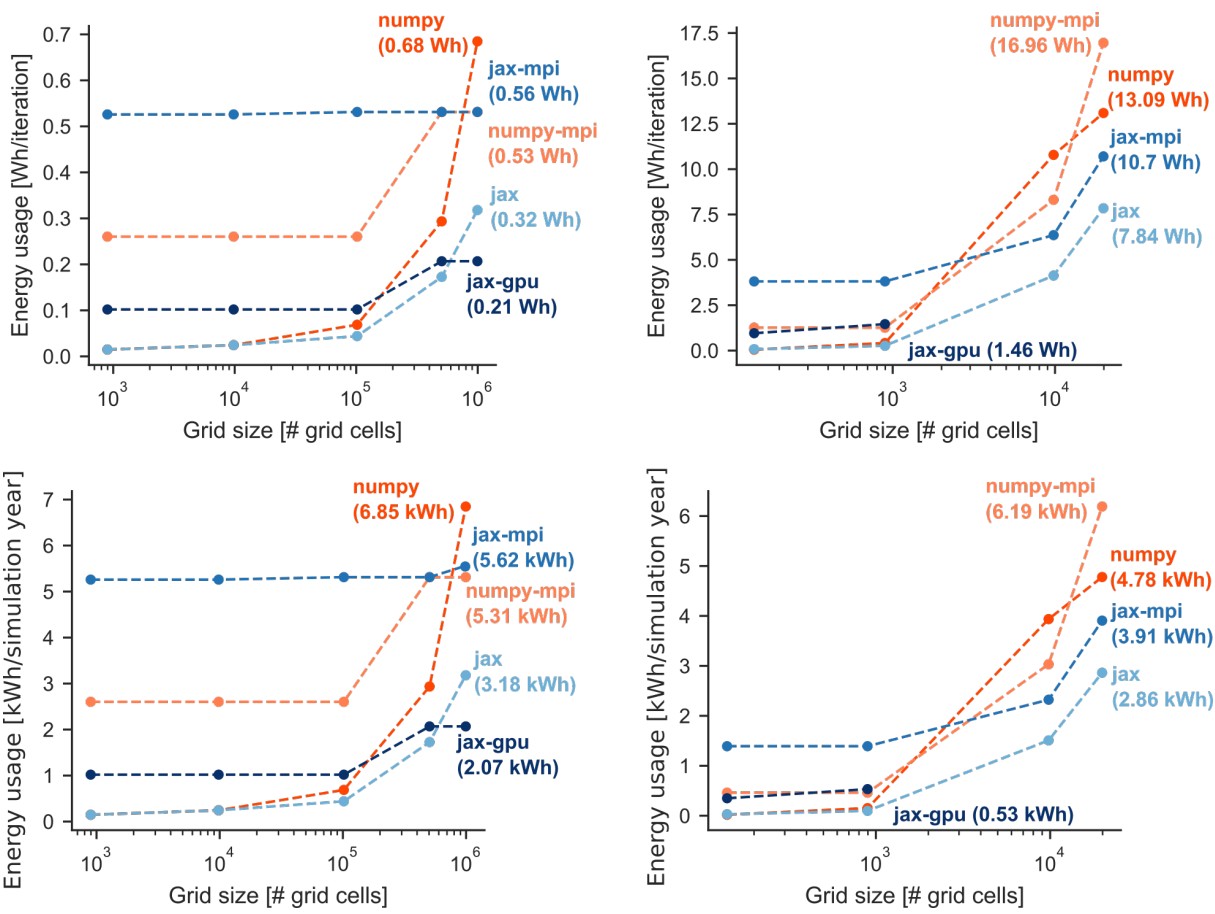

**Figure 4** Energy usage of computational back-ends on a cluster node for the RoGeR-SVAT type model (a, b) and for the RoGeR-SVAT-$^{18}$O type transport model (c, d)

HPC consumes more energy than running computations on a notebook. Depending on the energy source, HPC contributes differently to climate warming (Lannelongue et al., 2021). In order to raise awareness about the energy usage in HPC context and to provide information for a sustainable allocation of computational resources, we profiled the energy usage of RoGeR in an HPC context (see Table 2). Based on the profiling of computation time, we calculated the energy usage of the five back-ends using the method proposed by Lannelongue et al. (2021).

The results (Fig. 4) show that using multiple CPUs (numpy-mpi and jax-mpi) consumes more energy than other back-ends. Using a single GPU device decreases energy usage, while computation time still competes with multiple CPUs (cp. Fig. 3). For small and intermediate modelling experiments, single CPU (numpy and jax) back-ends use less energy than other back-ends. With these results, we aim to support efficient and sustainable allocation of computational resources. We suggest that computation time and energy usage should be considered

equally for the allocation.

## 4 Application: Soil water balance, $^{18}$O transport and water age statistics of a three-year period

To demonstrate the capabilities of RoGeR, we present a simple application example. We simulate the soil water balance and fluxes and $^{18}$O fluxes of the Eberbaechle catchment (1.54 km$^2$ with a resolution of 25 m x 25 m) for
a time period of three years. The input data was retrieved from the database WeatherDB, which provides data from stations operated by Deutscher Wetterdienst (DWD) tailored to the required format of RoGeR (Schmit, 2022). We

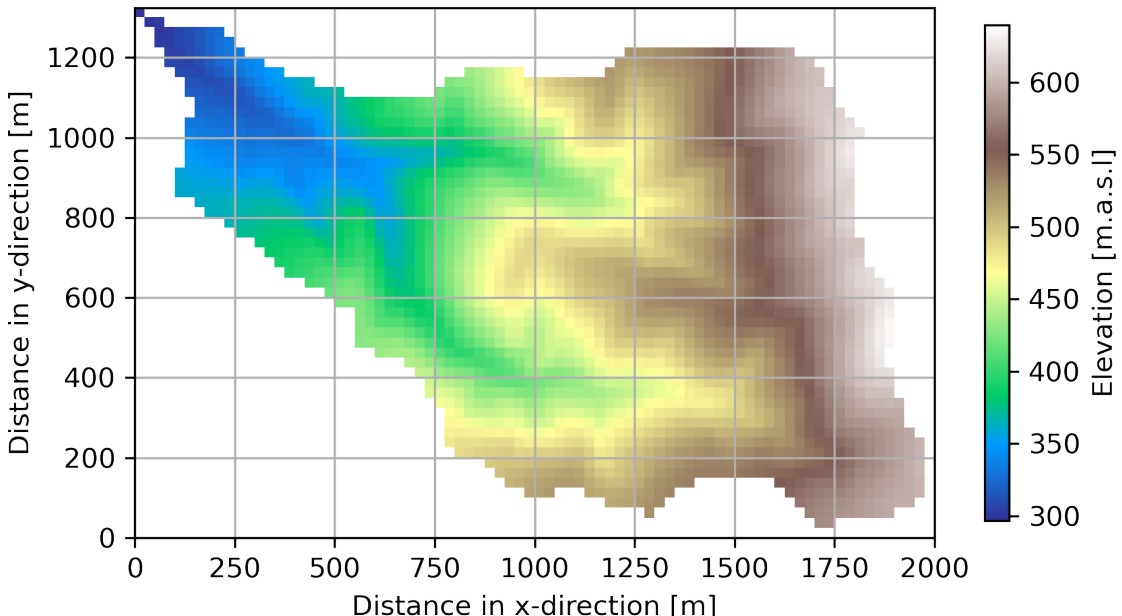

**Figure 5** The Eberbaechle catchment used for the application example. The Further catchment properties used as model parameters are shown in Figs. S28 – S39. The coordinates of the catchment outlet are 47°57'24"N 7°49'48"E.

selected the DWD station at Freiburg airport (station ID: 1443) to obtain precipitation, air temperature and potential evapotranspiration data from November 2019 to October 2022. Since DWD stations do not measure solute concentrations in precipitation, data for $^{18}$O in precipitation has been generated by a sinusoidal function
with random variation for amplitude and offset (Allen et al., 2018; amplitude=4.3±0.5 [‰], offset=-10±0.5 [‰] and phase=60 [days]). In order to set the values for the model parameters listed in Table 3, we used the soil map BK50 (Regierungspräsidium Freiburg, Landesamt für Geologie, Rohstoffe und Bergbau), LIDAR data (Landesvermessungsamt Baden-Würrtemberg), ATKIS DLM25 (Landesvermessungsamt Baden-Würrtemberg). Additionally, we assumed a deep groundwater table implemented through a high hydraulic conductivity of the
bedrock (see Table 3). SAS parameters for the selected power law distribution function are assumed to be spatially and temporally constant for each hydrological process and grid cell. We assigned k=0.2 to soil evaporation and

capillary rise, k=0.5 to transpiration, k=1.5 to percolation of root zone and k=1.5 to percolation of subsoil. Thus, soil evaporation capillary rise, and transpiration have a preference for younger water, while percolation processes have a preference for older water (see Fig. 2a).

**Table 3** Model parameters for the Eberbaechle catchment

| Hydrological model parameter | Symbol | Unit | Range of parameter values |
|---|---|---|---|
| Land use/Land cover | $lu\_id$ | - | Grass, forest, sealed surface, agriculture |
| Surface sealing | sealing | - | 0 - 1 |
| Soil depth | $z_{soil}$ | mm | 200 - 1000 |
| Length of vertical macropores | $l_{mpv}$ | mm | 0 - 800 |
| Density of vertical macropores | $\rho_{mpv}$ | $m^{-2}$ | 0 - 200 |
| Air capacity of soil | $\theta_{ac}$ | - | 0.06 - 0.14 |
| Plant available field capacity of soil | $\theta_{ufc}$ | - | 0.1 - 0.2 |
| Permanent wilting point of soil | $\theta_{pwp}$ | - | 0.09 - 0.18 |
| Saturated hydraulic conductivity of soil | $k_s$ | mm h$^{-1}$ | 10.4 - 125 |
| Hydraulic conductivity of bedrock | $k_f$ | mm h$^{-1}$ | 2500[a] |
| Offset for air temperature | $TA_{offset}$ | °C | -2.5 - -0.4 |
| Weighting factor for potential evapotranspiration | $PET_{weight}$ | - | 0.73 - 1.06 |
| Weighting factor for precipitation | $PRECIP_{weight}$ | - | 1.0 - 1.27 |

[a]parameter value represents free drainage. We assume free drainage since information for the lower boundary condition (i.e. depth of the groundwater table) is not available.

In Figure 6, we display the time series of hydrologic fluxes and soil water content with the corresponding $^{18}$O signature and water age distributions of a single grid cell. The temporal pattern exhibits that soil water content and travel times of hydrologic fluxes can be related. This pattern emphasizes the interlinkage between hydrologic

states and transport velocities of solutes (Hrachowitz et al., 2016). Figure 7 shows the cumulative distributions of soil hydrologic fluxes, soil water content, $\delta^{18}$O signals and average water ages at four different dates with different soil water content conditions. Soil water content is wetter at 10$^{th}$ February 2021 and drier at 13$^{th}$ August 2022 while the other two dates represent the transition between drier and wetter conditions. The cumulative distributions of $\delta^{18}$O signals and average water ages reveal differences for these different soil water content conditions. The

$\delta^{18}$O signals display distinct differences between the considered fluxes and soil water storage. The average water age exposes a more general pattern. For drier conditions, average water age is older, whereas for wetter conditions median water age decreases.

The primary objective of the example is to demonstrate the capabilities of RoGeR. Therefore, we kept the complexity of the example at a simple level. Although a comparison between simulations and observations is

important to evaluate the fidelity of the model, we do not provide such a comparison here. Instead, we refer to Schwemmle and Weiler (2023) for an in-depth evaluation of RoGeR using measurements from a grassland lysimeter site. Since the development of RoGeR as scientific software started recently and is still ongoing, further evaluation of RoGeR will be addressed in the future.


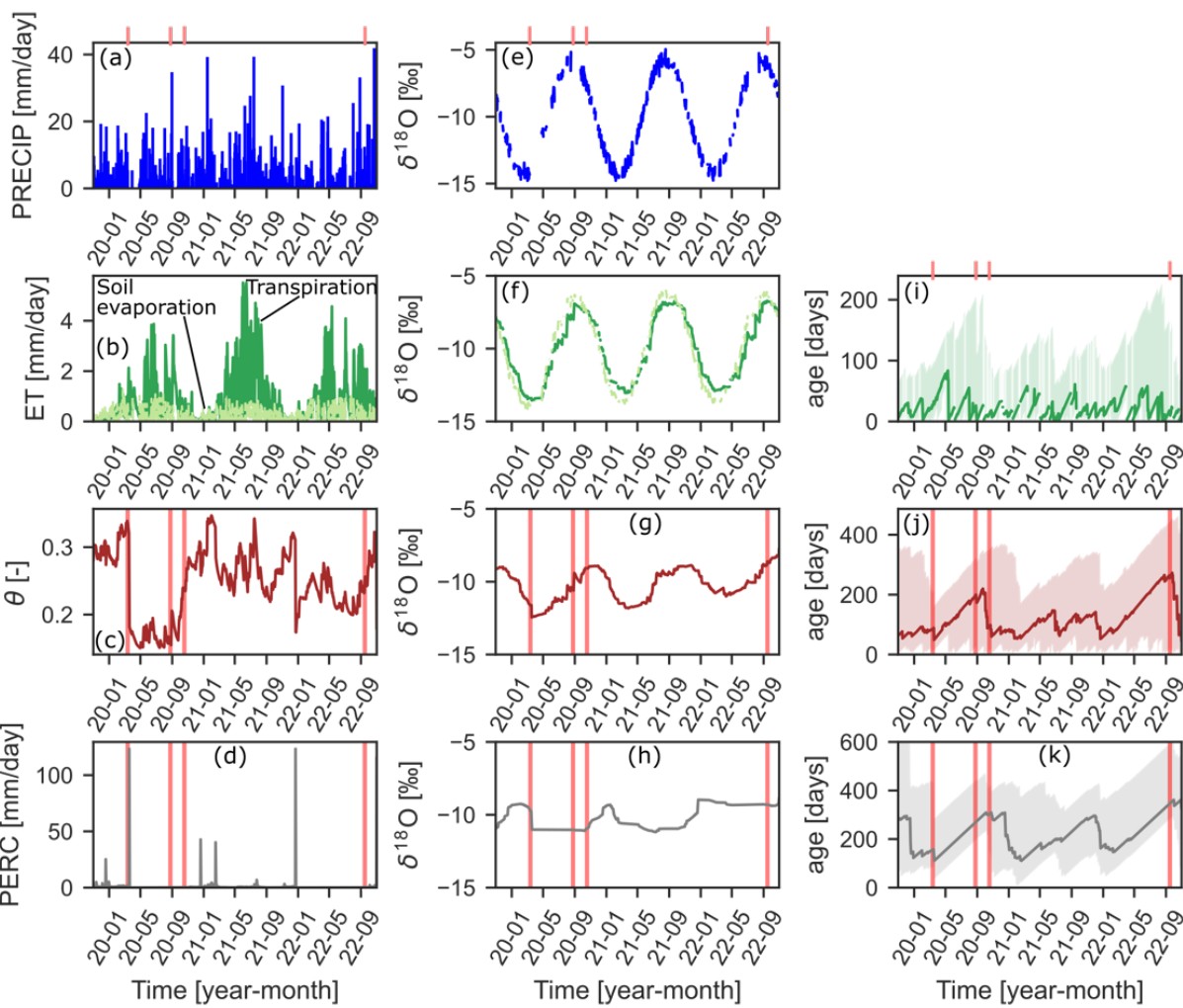

**Figure 6** Simulated fluxes and soil water content (a-d), corresponding $\delta^{18}O$ signal (e-h) and corresponding $10^{th}$, $50^{th}$ and $10^{th}$ percentile of water ages (i-k) of a single grid cell. Vertical red lines indicate the four different dates from Figure 6. Power law distribution function serves as SAS function (SAS parameters: $k_{evap\text{-}soil}$=0.2, $k_{transp}$=0.5, $k_{perc\text{-}rz}$=1.5, $k_{perc\text{-}ss}$=1.5; see Figure 2)

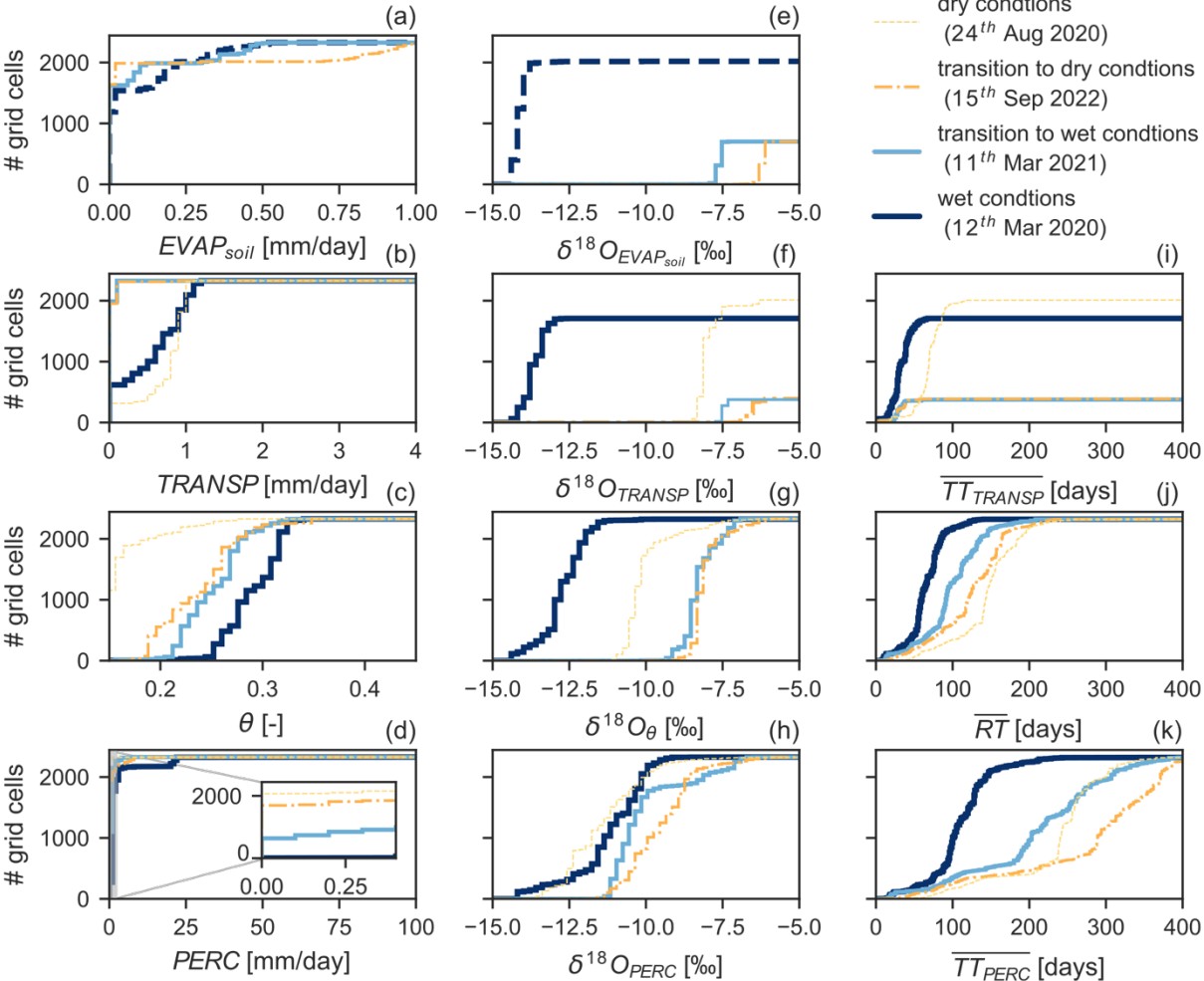

**Figure 7** Cumulative distributions of simulated fluxes and soil water content (a-d), corresponding $\delta^{18}$O signal (e-h) and the corresponding average travel time and average residence time (i-k) of the Eberbaechle catchment (1.54 km$^2$) at four different dates (transition to dry, dry, transition to wet and wet conditions)

The simple application example demonstrates the potential of RoGeR for a combined quantification of the water balance and solute mass balance. The example focusses on vertical soil hydrological processes and a conservative

tracer, but this is just an excerpt of the toolbox. Other processes (e.g. lateral subsurface runoff, different SAS function; see Sect. 2.2 and 2.3) or other tracers (e.g. bromide; see Sect. 2.3) could also be considered and implemented.

## 5 Summary and outlook

The development of the process-based hydrological toolbox RoGeR followed open-source software guidelines (Sect. 2.1). We believe that such guidelines improve the reproducibility in computational hydrology. With the modular code structure (Sect. 2.1.1) and the good readability of Python code, RoGeR is intended to be easy to use (i.e. usable by programmers with little experience) and to be easy to modify (i.e. modification and extension of the code). With using different computational back-ends, we maintained code readability without hampering computational performance (Sect. 2.1.2). The five back-ends provide the opportunity to simulate anything between plot scale and the catchment scale with reasonable computation times. Especially, the GPU back-end has great potential to reduce computation time and energy usage of catchment scale modelling experiments (Sect. 3). In comparison to the publicly available hydrological models written in Python, we combined hydrological processes (Sect. 2.2) and solute transport based on SAS (Sect. 2.3). The combined representation enables the prediction of hydrologic states and fluxes and their corresponding solute concentrations including travel times. The simple application example considering the water balance and $^{18}$O transport through the soil of the Eberbaechle catchment showed plausible results. The RoGeR toolbox contains many processes to describe one dimensional hydrological processes (i.e. no lateral transfer between grid cells). The implementation of the lateral transfer between grid cells (i.e. routing schemes for surface and subsurface runoff) will be addressed in future releases. Surface runoff routing will be implemented using a hydraulic scheme. Subsurface runoff routing will use the approach of Steinbrich et al. (2016) which is based on the topographic slope and corresponding flow velocities. Moreover, we suggest that future work may improve or extend the currently available process representations (e.g. gravity-driven infiltration and percolation; Demand and Weiler, 2021; Germann and Prasuhn, 2018) and further evaluation of RoGeR with measured data may provide insights on the strengths and weaknesses.

RoGeR contributes to a further diversification of the hydrological model landscape and the disagreement about process representation in the hydrological modelling community will continue (Weiler and Beven, 2015). In general, an advantage of this diversification and disagreement is that many different approaches are available and, hence, a great flexibility to address different problems. On the other hand, the theoretical diversification is accompanied by technical diversification (e.g. different programming languages or different data formats) that lead to inconsistencies in the application. We suggest that the diverse hydrological model landscape might benefit from focussing on constrained data interfaces of the models following common data conventions (Hallouin et al., 2022) and implementing standardized model interfaces (Hut et al., 2022; Hutton et al., 2020). This would facilitate the coupling of hydrological models with other models (e.g. groundwater model). Another advantage would be that multiple hydrological models could be compared more easily. Such a model comparison of RoGeR with other models, for example, with tRIBS (Ivanov et al., 2004a, b; Vivoni et al., 2004), DHSVM (Wigmosta et al., 1994)

or mHM (Samaniego et al., 2010) may be useful to highlight advantages and disadvantages of using RoGeR compared to other models.

*Supplement.* The supplement related to this article is available online at: https://doi.org/10.5194/gmd-2023-118-supplement


*Code availability.* The code is open-source and publicly available https://doi.org/10.5281/zenodo.10479343 and https://doi.org/10.5281/zenodo.8095094.

*Data availability.* The meteorological input data used in the application example has been retrieved from
https://weather.hydro.intra.uni-freiburg.de/ (Schmit, 2022) and is available at https://github.com/Hydrology-IFH/roger/tree/main/examples/hillslope_scale/svat_distributed_tutorial.

*Author contributions.* MW conceived the idea of the hydrological model RoGeR. MW, HL, AS and RS conceptualized RoGeR. HL and AS developed the first model suites of RoGeR in Python. RS developed RoGeR
as a software package with support from HL translating python code into a software package. RS wrote the first draft of the manuscript with contributions from all co-authors.

*Competing interests.* The authors declare that they have no conflict of interest.

*Acknowledgements.* The authors acknowledge support by the High Performance and Cloud Computing Group at the Zentrum für Datenverarbeitung of the University of Tübingen, the state of Baden-Württemberg through bwHPC and the German Research Foundation (DFG) through grant no INST 37/935-1 FUGG. We are grateful to the Veros development team for providing their software architecture, to the Python community for providing the useful tools and to everyone involved in the development of RoGeR.


*Financial support.* This research has been supported by Helmholtz Association of German Research Centres (grant no 42-2017). The article processing charge was funded by the Baden-Wuerttemberg Ministry of Science, Research and Art and the University of Freiburg in the funding programme Open Access Publishing.

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
