# Peer review of "$RoGeR\ v3.0.5 - a\ process-based\ hydrological\ toolbox\ model\ in\ Python$"

_Geoscientific Model Development, 2023_

## Referee Comment (RC2)

[referee-annotated manuscript omitted]

---

## Author Comment (AC1)

**Response to Reviewer #2**

We would like to thank the anonymous referee for his/her interest and the comments on our manuscript. Below, reviewer comments are in italic font and our replies are in plain blue font.

*This paper presents a python library to implement RoGer hydrological model. The main reason behind developing this library was to provide an-easy-to-use, reproducible and modular code. The library itself can be quite useful given that it also tries to simulate water isotope data.*

We thank the reviewer for his/her helpful comments.

*A major limitation of the paper is that it does not represent a real-world application of the code; we do not learn much by the synthetic example shown. The real challenge in implementing a distributed hydrological model is estimating the parameters that should be used for simulations, especially given the uncertainties in already limited hydrological data. Therefore, I suggest that the authors present a real-world case study in the paper. Also, a brief discussion of the calibration problem might benefit the paper; is there any plan to include calibration modules in the package?*

We present now a real-world application in chapter 4. Instead of calibrating the model, we used a soil map, LIDAR data and land cover date to derive the model parameters (see lines 424 ff.). Therefore, we do not discuss the model calibration. However, in the GitHub-Repository of the toolbox, we provide an example how model parameters can be estimated using a Monte Carlo approach.

*Another reason for the lack of reproducibility is a lack of calibration standards and lack of standards for dealing with uncertainties in hydrological modeling.*

We agree that such standards would help to improve reproducibility and include it in Chapter 1.

*This, in my opinion, might be a real advantage of RoGeR package. But one question that needs to be addressed is how good is this model compared to several other models available (tRIBS, VIC etc.)?*

A model comparison would be very interesting. However, the main focus of this manuscript is to describe and present the toolbox. We suggest, that such a model comparison should be addressed by another study. We add this suggestion to Chapter 5.

*Which routing scheme? There are many routing methods, so please be specific.*

We specified the routing schemes (see lines 534 ff.).

*Not quite clear. Is this the finest resolution the model accepts? If so, why? On the other hand, if this is the coarsest resolution, then the model is not useful for several watersheds around the world where precipitation is available as hourly or daily.*

The 10 minutes time step is required for the detailed representation of the runoff generation processes (i.e. infiltration, surface runoff and lateral subsurface runoff). Averaging the input flux for longer time steps leads to an overestimation of infiltration and underestimation of overland flow and preferential flow. Hourly precipitation or daily precipitation datasets can be used with the model and resampled to 10 minutes, however, losing the required temporal variability to correctly simulate the runoff generation processes. We added the justification for the input requirements to section 2.1.3 and refer to resampling methods to generate sub-daily precipitation from daily precipitation.

*But does this really ensure a physically meaningful quantities. The soil properties may vary at scales smaller than 25 m2.*

Here, we recommend a resolution for which we assume a consistent representation of the processes, but this can be different in different environments. And of course, soil properties may vary.

*Not necessarily regular. There are models where irregular grids are used.*

We describe the spatial discretization of RoGeR in section 2.1.3. Polygons can be used for RoGeR-1D (i.e. no lateral transfer between grid cells) as well. We clarified in the text that RoGeR models are meant.

*It is unclear whether this implementation of the model adds any new processes to the already existing model structure or not. Please clarify.*

*Define \omega and T.*

We added the missing definition.

*These variables should be italic.*

We changed the variables to italic.

*is it hydrologically sound? You should provide some guidance on distributing the model parameters? How convenient is it to distribute modeling parameters?*

*So, this is a hypothetical area? Why not show an application for a real small watershed?*

We show now a real-word application.

*Some guidance on how these parameters can be generated in real watershed application will be useful. Note that model calibration is perhaps the most difficult part in hydrological modeling, especially given the uncertainties in available hydrological data.*

We show now a real-word application.

*What is the reason for this assumption.*

Unfortunately, it is not clear to us which assumption is meant by this comment. We would like to ask the reviewer to refer to the lines in the manuscript. This would help us to locate the assumption-

*So, the colored areas bound the 25th and 75th percentiles.*

Yes, we changed now the percentiles to $10^{th}$ and $90^{th}$ percentiles.

*Write something about calibration of the model parameter in this section.*

Chapter 4 includes information how we derived the model parameters

*Why not do this in this paper itself. The example with artificial data is not very helpful.*

Again, we show now a real-word application.

*However, the important question is to figure out which model is good or bad for a given watershed. You have not discussed this problem at all in the paper. Why should a modeler choose RoGer over several other models?*

We agree that this question is important. However, the question is very watershed-specific and should be evaluated for each watershed individually. Moreover, a robust evaluation of the simulated variables requires the corresponding measurements. A robust evaluation of RoGeR has already been conducted in Schwemmle et al. (2023), Steinbrich et al. (2016) and Weiler (2005). Future work should address the evaluation of RoGeR at the catchment scale. We focused on the usability of the model to facilitate the usage by other modelers. Another reason for using RoGeR is that a little number of model parameters are required. Moreover, if environmental information is available, model parameters can be derived from such information instead deriving the model parameters by calibration.

---

## Author Comment (AC2)

**Response to Reviewer #1**

We would like to thank the anonymous referee for his/her interest and the comments on our manuscript. Below, reviewer comments are in italic font and our replies are in plain blue font.

*The authors present an interesting new hydrologic modelling toolbox in Python. The toolbox itself is very interesting and the publication is appropriate for GMD. I have a few points that the authors might want to address to improve their manuscript.*

We thank the reviewer for his/her helpful comments.

*(1) The toolbox itself is interesting and is mostly well described. What took me a while to understands though, is whether the toolbox models only described 1-D soil processes or whether they can present a grid with spatial variability. As well as how this raster would be connected. My suggestion is that the authors expand Figure 1 to also include spatial and temporal discretization, as well the inputs to the model. I think it would help the reader to have such a conceptual figure regarding the possible model architectures.*

We expanded Figure 1 and added the input(s), temporal and spatial discretization to the figure. Currently, the toolbox describes only 1-D processes. The implementation of the lateral fluxes between the grid cell is planned for the future as already mentioned in lines 295 ff.

*(2) A related aspect is the very specific input requirements for the model discussed in lines 112ff. I am a bit confused why this is so specific when the model architecture is meant to be modular. Should the model not be adjustable to the data available? Precipitation of 10 minute time steps or less will limit available datasets very much. Why can the model not be run with hourly time steps? Especially given that other inputs are daily. I think the authors should provide some justification here, rather than just stating this as a fact.*

The 10 minutes time step is required for the detailed representation of the runoff generation processes (i.e. infiltration, surface runoff and lateral subsurface runoff). Especially high rainfall intensities may cause infiltration excess which leads to Hortonian surface runoff that cannot be adequately represented by hourly time steps. Hourly precipitation or daily precipitation datasets can be resampled to 10 minutes. We added the justification for the input requirements to section 2.1.3.

*(3) Another part where I would expect a bit more discussion is section 2.2. Much has been published on the issue of modular hydrologic models over the last 20 years, but hardly anything from this discussion is mentioned here (e.g. https://gmd.copernicus.org/articles/12/2463/2019/gmd-12-2463-2019.pdf). This discussion should also include how a certain process module (e.g. transpiration or surface runoff) could be represented using different process complexities. I assume this is included here.*

The main reason for using a modular structure was to improve the readability of the code. Of course, the different processes can be represented by different degrees of complexity but this would lead to multiple modules for a single process and increase the choice of modules. Modular means that certain processes (e.g. lateral subsurface runoff, capillary rise from groundwater, etc.) can be activated/deactivated. Although this provides a high flexibility, the combination of multiple modules into a process consistent model can be challenging for model users. Therefore, we provide the processes with a certain complexity and hence limit the flexibility of the model. However, by providing pre-defined models we intend to guarantee process consistent combination of the modules. We include the mentioned reference and further discuss the modular usage in the context of the mentioned reference in section 2.2.

*(4) Could the authors expand a bit on the numerical implementation of the models and how this links to previous discussions on the topic (https://agupubs.onlinelibrary.wiley.com/doi/full/10.1029/2009WR008894)?*

In order to calculate the soil hydrological fluxes and soil water content, we use analytical solutions of the infiltration into soil matrix and macropores (see Steinbrich et al., 2016 and Weiler, 2003) and variable time stepping which

depends on the rainfall intensity, but numerical implementations are not necessary. For the solute transport, three different numerical schemes are available and described in section 2.3. We add the information about the numerical implementation to section 2.2 and discuss it in the context of the mentioned reference.

*(5) In line 248, the authors mention that parameters are randomly generated, but do not explain more. I assume the authors sampled from independent uniform distributions. Are there other options than this approach? Here the link to other toolboxes might be particularly useful to further explore the model outputs given that more and more analysis algorithms can be run with generic input samples (e.g. https://www.sciencedirect.com/science/article/abs/pii/S1364815218303220).*

We modified Chapter 4 and run the simulations for a real-world catchment rather than using randomly generated data. The mentioned toolbox might be interesting for a more advanced analysis of the simulations. Generally, the toolbox can be used with other toolboxes. For this study, such an advanced analysis is beyond the scope of a model description paper.

*(6) If the model is represented as a grid, is there a possibility for river flow actually occurring? Or is the domain limited to plot scales where channelized flow might not occur?*

Currently, river/channel flow has not been implemented, yet. The implementation of river/channel flow is planned for the future as already mentioned in lines 295 ff.

*(7) A hydrologic modelling toolbox like the one presented here can (and probably should) link to a wider range of Python toolboxes that assess and attribute uncertainties (e.g. https://safetoolbox.github.io/) or estimate parameters (e.g. https://pymoo.org/). Discussing how one might use existing toolboxes with RoGeR would be interesting for users I believe.*

In the GitHub-Repository of the toolbox, we provide some examples how RoGeR can be used with other toolboxes. We provide examples that estimate parameters using a Monte Carlo approach or analyses the parameter sensitivities (https://salib.readthedocs.io/en/latest/index.html).

*(8) There are a few issues with the writing. E.g. "measurements for their states" should be measurements of, the last sentence in the abstracts includes and twice and is too long, ... So please have another read through just for grammar.*

We had another read and improved the grammar.

---

## Author Response (AR2)

**Response to Reviewer #1**

We would like to thank the anonymous referee for his/her interest and the comments on our manuscript. Below, reviewer comments are in italic font and our replies are in plain blue font.

*The authors improved the manuscript from the previous version. However, I still struggle with a lack of clarity regarding what the toolbox and the models it can build are like. The choice of comparable models and applications seems – to me – rather different from the details of the model described by the authors. Could the authors try to still be clearer in this regard? I am sorry if this is annoying the authors, but I believe more clarity will help the readers.*

We thank the reviewer for his/her helpful comments. However, we are a bit surprised, that the reviewer came up with completely new points, he/she did not mention in the first round.

*(1) The authors still do a poor job in describing what the toolbox is for. If the toolbox is meant to run with forcing data in the order of minutes and spatial resolutions up to 25m2, then clearly the objective is not to build continental scale models. However, one must read significantly into section 2 before this becomes apparent. The example models used in the introduction section vary widely in model style (e.g. HYPE and HYDRUS). I think that the proposed model is a valuable addition to the current set of models by adding a very high resolution and small-scale model to the mix. Why not say so? Why do the authors not define what the model is for (and what it is not for)?*

We clarify the description of the toolbox and elaborate the scale of application. We added the missing information about the model objectives to the introduction. We further stress the value added by RoGeR in the introduction. In lines 169f., we provide guidance on the spatial resolution. This should avoid potential misuse. Moreover, capabilities of the models are listed in section 2.2 and section 2.3. With the information about the considered processes and temporal resolution, any hydrologist should be able to judge if the model is suitable for a certain application he/she has in mind.

*(2) The current motivation (in the abstract) suggests that the model is meant to improve water management. However, the model needs very high-resolution data on forcing (<10 minutes) and system properties (<25m2). This is not what a typical hydrological model for water management uses, even if it does include water quality. The model is much more like DHSVM [a] than to HYPE or other models used in the text. Instead, the model select the models they include by the language they have been written in, which is less helpful to understand what this specific model actually does.*
[a] https://www.pnnl.gov/projects/distributed-hydrology-soil-vegetation-model

We modified the motivation in the abstract and refined the scale of the water management. We already provide a brief description of the capabilities of HYDRUS, EcH2O-iso and mHM in the beginning of the paragraph (see lines 35ff.). We include the programming language since knowledge in certain programming plays an important role for the application of the considered model.

*(3) In lines 31ff. („The main reasons for this lack of reproducibility…“) you cite Reinecke et al. (2022) several times in a single sentence. It is perfectly fine to just cite it once at the end of the sentence.*

We merged the citation and moved the citation to the end of the sentence.

*(4) The authors state in their response that "Modular means that certain processes (e.g. lateral subsurface runoff, capillary rise from groundwater, etc.) can be activated/deactivated." That is a unique definition used by the authors and rather different from the typical use of the term modular in the context of model building in hydrology, which is to define the option to represent the same process with multiple algorithms (e.g. Knoben et al. 2019, GMD). I would at a minimum expect that the authors clarify that their use of the term modular is different from the more common use. Though preferably the authors use a different and less confusing terminology.*

Knoben et al. (2019) use the following definition of modularity:
"MARRMoT takes inspiration from earlier modular frameworks (e.g. FUSE, Clark et al., 2008; FLEX, Fenicia et al., 2011) and uses modular code with individual flux equations as the basic building blocks. Multi-model frameworks benefit from modular implementation because this simplifies the programming of the framework and makes it easier to (i) reuse components of a model in a different context, including cases in which the same basic equation is used by multiple models, and (ii) add new options to the framework."
Similarly, we provide the flux equations in process-specific modules that are used as basic building blocks. Therefore, we argue that we do not deviate from definition.

*(5) I am afraid that I am unclear about the organization proposed in table 1. Software complexity from low to high is used to organize guidelines for open software. I do like the guidelines, but I do not see how these points organize along a complexity axis. Online documentation or unit tests are only useful to high complexity software? I think unit tests should be done for any software of the type proposed here. If the authors keep the complexity organization, then they need to explain why these guidelines follow this logic.*

We further explain the rationale behind the complexity organization in Section 2.1.

*(6) The authors conclude by linking their model to others like VIC and the potential for integration with Earth Science Models. However, models like VIC and certainly Earth Science Models run at spatial scales of kilometers, where 1km2 resolution is considered hyper resolution. Your model runs at meters. Do the authors really suggest that VIC is the right model to compare their model against? I mentioned DHSVM earlier. DHSVM and VIC come from the same group but have been built to fulfill very different purposes. The model the authors describe is very much like DHSVM, so why do the authors compare it to VIC?*

We removed the potential integration into earth science models. Instead, we link towards potential model coupling opportunities. We also removed the comparison to VIC und use DHSVM instead.

**Response to Reviewer #2**

We would like to thank the anonymous referee for his/her interest and the comments on our manuscript. Below, reviewer comments are in italic font and our replies are in plain blue font.

*Authors have addressed all of my comments now. Thank you for including a real-world example in the paper, even though, it would have been better to use a watershed where real-world O18 were available. Also, I was referring to the assumption made about the bedrock conductivity in the paper. It would be nice to see a rationale for using a high value. Finally, some information about the study catchments such as climatological and hydrological conditions, soil properties and geology would be nice to include.*

*These are just some suggestions that the author may address. Otherwise, I recommend the publication of the paper.*

We thank the reviewer for his/her helpful comments.

The rationale for using a high bedrock conductivity is the assumption of free drainage. We are assuming free drainage since knowledge about the lower boundary condition (i.e. the depth of the groundwater table) is not available. We clarify the assumption in Section 4. Regarding information about climatology, soil properties and land cover, we would like to refer to Section S3 in the supplement.